# Train-by-Reconnect: Decoupling Locations of Weights from Their Values

**Yushi Qiu**      **Reiji Suda**
Graduate School of Information Science and Technology, The University of Tokyo
{yushi621, reiji}@is.s.u-tokyo.ac.jp

## Abstract

What makes untrained deep neural networks (DNNs) different from the trained performant ones? By zooming into the weights in well-trained DNNs, we found that it is the *location* of weights that holds most of the information encoded by the training. Motivated by this observation, we hypothesized that weights in DNNs trained using stochastic gradient-based methods can be separated into two dimensions: the location of weights, and their exact values. To assess our hypothesis, we propose a novel method called *lookahead permutation* (LaPerm) to train DNNs by reconnecting the weights. We empirically demonstrate LaPerm's versatility while producing extensive evidence to support our hypothesis: when the initial weights are random and dense, our method demonstrates speed and performance similar to or better than that of regular optimizers, e.g., *Adam*. When the initial weights are random and sparse (many zeros), our method changes the way neurons connect, achieving accuracy comparable to that of a well-trained dense network. When the initial weights share a single value, our method finds a weight agnostic neural network with far-better-than-chance accuracy.

## 1  Introduction

Conventional gradient-based algorithms for training deep neural networks (DNNs), such as stochastic gradient descent (SGD), find the appropriate numerical values for a set of predetermined weight vectors $\theta$. These algorithms apply the changes $\Delta\theta$ to $\theta$ at each iteration. Denoting the weight vectors at the $t$-th iteration as $\theta_t$, the following update rule is used: $\theta_t \leftarrow \theta_{t-1} + \Delta\theta_{t-1}$. Therefore, we have the relationship between a trained and an untrained DNN: $\theta_T = \theta_0 + \sum_{t=1}^{T} \Delta\theta_t$, given the initial weights $\theta_0$ and the weights $\theta_T$ obtained by training the network for $T$ iterations. However, since $\Delta\theta_t$ is dependent on $\theta_{t-1}$ and $\Delta\theta_{t-1}$ for every $t$, it is difficult to directly interpret from the term $\sum_{t=1}^{T} \Delta\theta_t$ what is the most substantial change that the training has applied to the initial weights.

In this work, we examine the relationship between $\theta_0$ and $\theta_T$ from a novel perspective by hypothesizing that weights can be decoupled into two dimensions: the *locations* of weights and their *exact values*. DNNs can be trained following the same stochastic gradient-based regime but using a fundamentally different update rule: $\theta_t \leftarrow \sigma_t(\theta_{t-1})$, for a permutation operation $\sigma_t$. Consequently, we have $\theta_T = \sigma_T(...(\sigma_1(\theta_0)))$, and thus have $\theta_T = \sigma_k(\theta_0)$ for $\sigma_k$ from the same permutation group. Supporting our hypothesis, in the first half of this paper, we demonstrate that SGD encodes information to DNNs in the way their weights are connected. In the latter half of this paper, we show that given an appropriately chosen neural architecture initialized with *random weights*, while fine-tuning of the exact values of weights is essential for reaching state-of-the-art results, properly determining the location of weights alone plays a crucial role in making neural networks performant.

In Section 2, we describe an interesting phenomenon in the distribution of weights in trained DNNs, which casts light on understanding how training encodes information. In Section 3, we show that

it is possible to translate stochastic gradient updates into *permutations*. In Section 4, we propose a novel algorithm, named *lookahead permutation (LaPerm)*, to effectively train DNNs by reconnection. In Section 5, we showcase LaPerm's versatility in both training and pruning. We then improve our hypothesis based on empirical evidence.

## 2  Similarity of Weight Profiles (SoWP)

DNNs perform chains of mathematical transformations from their input to output layers. At the core of these transformations is *feature extraction*. Artificial neurons, the elementary vector-to-scalar functions in neural networks, are where feature extraction takes place. We represent the incoming weighted connections of a neuron using a one-dimensional vector (flattened if it has a higher dimension, e.g., convolutional kernel), which we refer to as a *weight vector*. We then represent all neuron connections between two layers by a *weight matrix* in which columns are weight vectors.

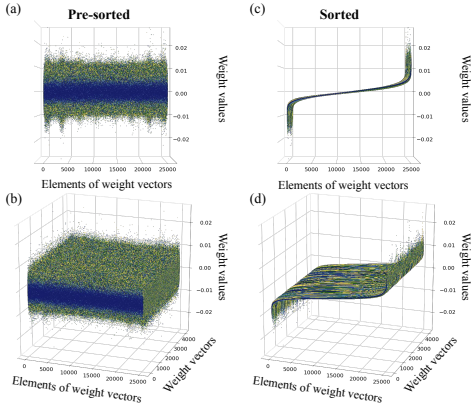

Figure 1: *Profiling* a weight matrix in a pre-trained VGG16 on ImageNet. Parts (b) and (d) are the same plots as (a) and (c), respectively, from different viewing angles. The color of each single scatter plot is chosen in order, cyclically, from midnight-blue, gold, dark-green, and steel-blue.

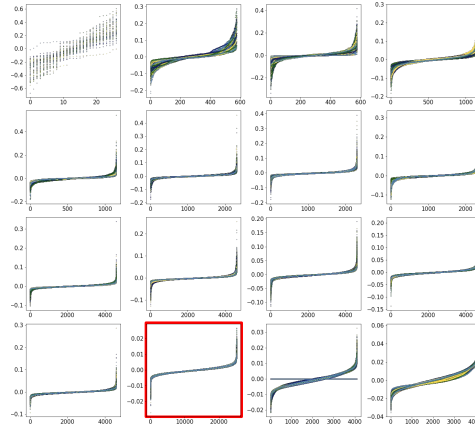

Figure 2: *Profiling* all weight matrices in a pre-trained VGG16. The weight profile selected for Figure 1 is marked in red. The *z*-axes are hidden, as in Figure 1.

To gain insight into how the information is encoded in a trained DNN, we zoom into the weight vectors. Here, we visualize a weight matrix by drawing a scatter plot for each of its weight vectors, where the *x*- and *y*-axis indicate the indices and weight values associated with each entry, respectively. We stack these scatter plots on the same figure along the *z*-axis, such that all plots share the same *x* and *y*-axis. Figure 1 (a) and (b) show such a visualization of a weight matrix from different viewing angles; it represents all weighted connections between the last convolutional layer and the first fully-connected layer in VGG16 [43] pre-trained on ImageNet [7]. At a glance, these plots appear to be roughly zero-mean, but as a jumble of random-looking points. Nothing is particularly noticeable *until we sort all the weight vectors* and redo the plots to obtain Figure 1 (c) and (d). The patterns shown on these figures imply that *all 4096 weight vectors have almost identical distributions and centers* as their scatter plots closely overlap with each other. In the forthcoming discussion, we refer to sorted weight vectors as a *weight profile*.

As shown in Figure 2, although the shapes may vary, similar patterns are observed in most layers. We also found these patterns in every other tested pre-trained convolutional neural network (CNN), such as ResNet50 [15], MobileNet [17], and NASNet [54]. Please refer to Appendix A.2 for their visualizations. We call this phenomenon of weight vectors associated with a group of neurons possessing strikingly similar statistical properties, the *similarity of weight profiles (SoWP)*. It reveals the beauty and simplicity of how well-trained neural networks extract and store information.

*Similarity* is where two or more objects lack differentiating features. SoWP implies that many features encoded by training are lost after sorting. In other words, *the features are mostly stored in the order of weights before sorting*. Consequently, SoWP allows the weights of a well-trained DNN

to be near-perfectly separated into two components: the *locations* (relative orders) of weights and a statistical distribution describing their *exact values*.

## 3   Is Permutation the Essence of Learning?

If information can truly be encoded in the relative order of weights, we would expect changes in this order to reflect the training progress. We train a fully-connected DNN with two hidden layers (100 ReLU units each) using the MNIST database [26]. In isolation, we train using the same architecture and initialization under three different settings: (1) SGD (1e-1) with no regularization. (2) *Adam* (1e-3) with no regularization. (3) *Adam* (1e-3) with L2 regularization [24]. The learning rates in all experiments are divided by 2 and 5 at the 10th and 20th epochs. Full experimental settings are given in the Appendix. We extract the orders of weights during the training by focusing on their *rankings* within each weight vector.

The *ranking* of a weight vector $w_j$ is defined as a vector $R_j$, where $\#R_j = \#w_j$, of distinct integers in $[0, \#w_j)$, such that $w_j[p] > w_j[q]$ implies $R_j[p] > R_j[q]$, for all integers $p, q \in [0, \#w_j)$, $p \neq q$. Here, $\#w_j$ denotes the number of elements in $w_j$; and $w_j[p]$ denotes the $p$-th element of the vector $w_j$. If the ranking $R_{j,t}$ of $w_j$ at the $t$-th iteration is different from $R_{j,t-1}$ at the $t-1$-th iteration, there must exist a *permutation* $\sigma_t$ such that $R_{j,t} = \sigma_t(R_{j,t-1})$. For simplicity, we compute $D_{j,t} = |R_{j,t} - R_{j,t-1}|$, which we refer to as the *ranking distance*. The $i$-th entry of the ranking distance $D_{j,t}[i]$ indicates the distance of change in the ranking of $w_j[i]$ in the past iteration.

For a weight matrix $W$, we compute mean: $\overline{D_t} = \sum_j \sum_i D_{j,t}[i]/\#W$, where $\#W$ is the total number of entries in $W$, and the standard deviation: $\mathrm{SD}[D_t] = \sqrt{\sum_j \sum_i (D_{j,t}[i] - \overline{D_t})^2/\#W}$ of the ranking distance.

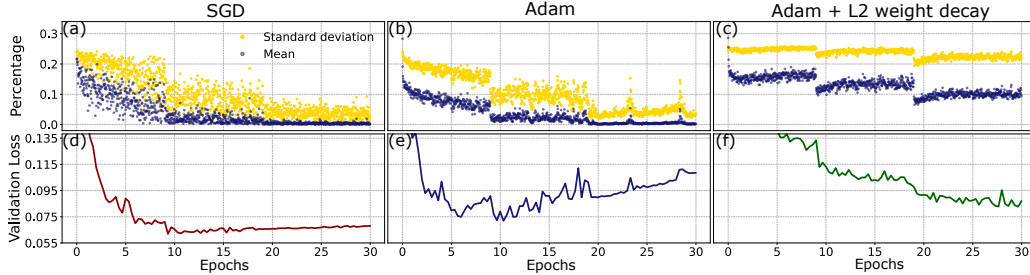

Figure 3: Monitoring ranking distance and validation loss in the first weight matrix of the network. Each column title indicates the experimental setting. Shown under each title (top to bottom) is the evolution of the ratio of the mean and standard deviation of the ranking distance to the size of the weight vector, i.e., $\overline{D_t}/784$ and $\mathrm{SD}[D_t]/784$, and the trend of validation loss on 10,000 test images.

**Results and Analysis**   We briefly point out that in the results shown in Figure 3, changes in the ranking reflect the progress of learning. The behaviors of permutations in (a)~(c) show unique traits under each setting. In (a), the trend seems random, especially when the learning rate is 0.1, reflecting how SGD updates largely depend on the randomly sampled batches. In contrast, in (b) and (c), since the *Adam* updates consider previous gradients, the permutations appear to follow a particular trend. In (c), when L2 regularization is applied, the change in ranking is more significant in both number and size. This implies that the weights become smaller and closer to each other because of the weight penalties. The closer they are, the easier their rankings can be swapped, and the greater the ranking distance the swap would cause by an update. Moreover, in (b) and (e) at around the 24th and 29th epoch, the sharp rise in the mean of the ranking distance predicts a deterioration in validation loss. The full experiment and analysis are presented in the Appendix.

## 4   Lookahead Permutation (LaPerm)

Motivated by the observation in Section 3 that the changes in the ranking (order) of weights reflect the progress of training, we try to achieve the inverse: we propose LaPerm, a method for training DNNs by *reconnecting* the weights. This method adopts an inner loop structure similar to the Lookahead (LA) optimizer [52] and the Reptile optimizer [38]. Pseudocode for LaPerm is shown in Algorithm 1.

We consider training a feedforward network $F_{\theta_0}(x)$ with initial weights $\theta_0 \sim D_\theta$. Before the training starts, LaPerm creates a copy of $\theta_0$ and sorts every weight vector of this copy in ascending order.

We store the newly created copy as $\theta_{\text{sorted}}$. At any step $t$ during training, LaPerm holds onto both $\theta_{\text{sorted}}$ and $\theta_t$, where $\theta_{\text{sorted}}$ is served as a preparation for *synchronization* and is maintained as sorted throughout training; Weights $\theta_t$, which are updated regularly at each mini-batch using an inner optimizer, Opt, of choice, e.g., *Adam*, are used as a reference to permute $\theta_{\text{sorted}}$.

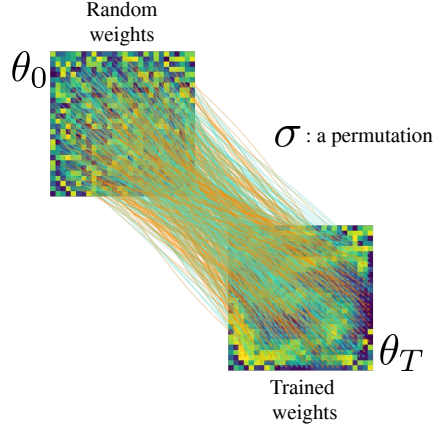

---

**Algorithm 1** LaPerm

**Require:** Loss function $L$
**Require:** initial weights $\theta_0$
**Require:** Synchronization period $k$
**Require:** Inner optimizer Opt
   $\theta_{\text{sorted}} \leftarrow$ Sort weight vectors in $\theta_0$
   **for** $t = 1, 2, \ldots$ **do**
      Sample mini-batch $d_t \sim D_{\text{train}}$
      $\theta_t \leftarrow \theta_{t-1} + \text{Opt}(L, \theta_{t-1}, d_t)$
      **if** $k$ divides $t$ **then**
         $\theta_t \leftarrow \sigma_{\theta_t}(\theta_{\text{sorted}})$ // Synchronization
      **end if**
   **end for**

---

Figure 4: (Left) Pseudocode for LaPerm. (Right) Given a randomly initialized fully-connected DNN with one hidden layer trained using MNIST [26], one typical weight vector associated with the hidden layer before (upper left) and after training (lower right) using LaPerm. Reconnected weights with same values are connected using a green or orange arrow chosen at random.

**Synchronization** Once every $k$ steps, *synchronization*: $\theta_t \leftarrow \sigma_{\theta_t}(\theta_{\text{sorted}})$ is performed, where $\sigma_{\theta_t}$ is a permutation operation generated based on $\theta_t$. We refer to $k$ as the *synchronization period* (sync period). More formally, synchronization involves the following two steps: 1) permuting the weight vector $w'_j$ in $\theta_{\text{sorted}}$ according to its counterpart $w_j$ in $\theta_t$ such that $w_j$ and $w'_j$ have the same *ranking* (defined in Section 3) for every $j$; and 2) assigning the permuted $\theta_{\text{sorted}}$ to $\theta_t$. It is important to keep weight vectors in $\theta_{\text{sorted}}$ as always sorted so that the permutation can be directly generated by indexing each $w'_j$ using the ranking $R_j$ of $w_j$ with no extra computational overhead. Optionally, we could make a copy $\theta'_{\text{sorted}}$ before synchronization and only permute $\theta'_{\text{sorted}}$ so that $\theta_{\text{sorted}}$ is unchanged.

In essence, how exactly the magnitude of weights in $\theta_t$ have been updated by Opt is not of interest; $\theta_t$ is only considered to be a correction to the ranking of $\theta_0$. In other words, we extract permutations from $\theta_t$. If the total number of training batches $N$ and synchronization period $k$ are chosen such that $k$ divides $N$, at the end of the training, the network's weights $\theta_T$ is guaranteed to be $\sigma(\theta_0)$, for a weight vector–wise permutation $\sigma$. A visualization of such permutation is shown in Figure 4 (Right).

**Computational Complexity** Sorting is required to get the rankings of weight vectors at synchronization. Suppose we use a linearithmic sorting method for weight vectors of size $\#w_j$, an inner optimizer with time complexity $T$, and sync period $k$. In this case, the amortized computational complexity for one LaPerm update is $O(T + \frac{1}{k}\sum_j \#w_j \log \#w_j)$. When $k$ and the learning rate of the inner optimizer are chosen such that the weight distribution and range of $\theta_t$ are similar to those of the initial weights $\theta_0$, the performance of sorting can be improved by adopting, e.g., bucket sort, especially when the weights are near-uniformly distributed. In modern DNN architectures, the average size of weight vectors is usually under $10^4$, e.g., in ResNet50 and MobileNet, it is approximately 1017 and 1809, respectively.

## 5 Experiments: A Train-by-Reconnect Approach

In this section, we reconnect randomly weighted CNNs listed in Table 1 trained with the MNIST [26] and CIFAR-10 [23] datasets using LaPerm under various settings. LaPerm has two hyperparameters to itself: the initial weights $\theta_0$ and sync period $k$. In Section 5.1, we examine how the distribution of $\theta_0$ affects LaPerm's performance. In Section 5.2, we vary the size of $k$ within a wide range and analyze its effect on optimization. In Section 5.3, based on the experimental results, we improve our hypothesis initially stated in Section 1. In Section 5.4, we test our hypothesis as well as comprehensively assess the capability of LaPerm to train sparsely-initialized neural networks from scratch. In Section 5.5, we create weight agnostic neural networks with LaPerm. Here, we only show hyperparameter settings

| Network | Conv7 | Conv2 | Conv4 | Conv13 | ResNet50 |
|---|---|---|---|---|---|
| Conv Layers | 2x32, 32(5x5;Stride 2)<br>2x64, 64(5x5;Stride 2)<br>128 (4x4) | 2x64, pool | 2x64, pool<br>2x128, pool | 2x64, pool, 2x128, pool<br>3x256, pool<br>3x512, pool, 3x512, pool | 16, 16x16<br>16x32<br>16x64 |
| FC Layers | 10 | 256, 256, 10 | 256, 256, 10 | 512, 10 | avg-pool, 10 |
| All / Conv Weights | 325k / 326k | 4.3M / 38K | 2.4M / 260K | 14.9M / 14.7M | 760K / 760K |
| Epochs / Batch | 45 / 50 | 125 / 50 | 90 or 125 / 50 | 125 / 50 | 200 / 50 |

Table 1: Architectures used in the experiments. The table is modified based on [9, 53]. Convolutional networks, if not specified, use 3x3 filters in convolutional (Conv) layers with 2x2 maxpooling (*pool*) followed by fully-connected (FC) layers. Conv2 and Conv4 are identical to those introduced in [9]. Newly introduced Conv7 is modified based on LeNet5 [25], and Conv13 is a modified VGG [43] network for CIFAR-10 adapted from [35].

necessary for understanding the experiments. Detailed settings are in Table 1 and Appendix A.4. The usage of batch normalization (BN) [19] is explained in Appendix A.4.6.

**5.1 Varying the Initial Weights** We train Conv7 on MNIST using different random initializations: He's uniform $U_H$ and normal $N_H$ [14], Glorot's uniform $U_G$ and normal $N_G$ [11]. We also train the same network initialized with $N_H$ using *Adam* [22] and LA [52] in isolation. The trained weights obtained with *Adam* at the best accuracy are shuffled weight vector–wise and used as another initialization for LaPerm, which we refer to as $N_S$. We use five random seeds and train the network for 45 epochs with a batch size of 50 and a learning rate decay of 0.95. We choose $k = 20$ for LaPerm. For all experiments in this paper, LaPerm and LA use *Adam* as the inner optimizer.

The results are presented in Figure 5. While only a small discrepancy is observed between LaPerm's validation accuracy using $U_H$ and $N_H$, they are consistently ~0.1% above that of $U_G$ and $N_G$, which shows the importance of the statistical properties of weight for LaPerm to reach the last bit of accuracy. $N_S$ performs similarly to $N_G$ until the 25th epoch, where it stops improving. A possible cause is that *Adam* over-adapted the values of weights in its training. When weights were shuffled to obtain $N_S$, it became difficult to rediscover the right permutation.

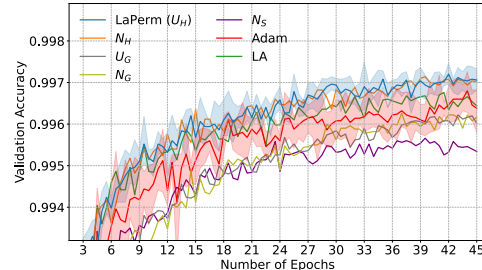

Figure 5: MNIST experiments. Text "LaPerm" is omitted except for $U_H$. The band, if shown, indicates the minimum and maximum values obtained from five runs, otherwise they are omitted for visual clarity.

Overall, although LaPerm pulls all the weights back to uniform random values every 20 batches ($k$=20), we see no disadvantage in its performance compared with *Adam* and LA. This observation implies that *the inner optimizer of LaPerm, between each synchronization, encodes information to the weights in a manner that can be almost perfectly captured by extracting their change in ranking*. In the end, we obtained state-of-the-art accuracy using MNIST, i.e., ~0.24% test error, which slightly outperformed Adam and LA.

**5.2 Understanding the Sync Period k** In Section 5.1, we observe that LaPerm succeeds at interpreting $\theta_t$ as a permutation of $\theta_0$. What happens if we vary the value of $k$? We train Conv4 on CIFAR-10 and sweep over 1 to 2000 for $k$. The results are shown in Figure 6(a). We observe an unambiguous positive correlation between the size of the sync period $k$ and the final validation and training accuracy. Interestingly, when $k = 2000$, i.e., sync only once every two epochs, its accuracy started as the slowest but converged to the highest point, whereas for $k \leq 100$, the trend starts fast but ends up with much lower accuracy. To see this clearly, in Figure 6 (b) and (c), we smoothed [42] the accuracy curves and found that before the 60th epoch (shown in (b)), the accuracies are negatively correlated with $k$, but are reversed afterward (shown in (c)).

It is worth noting that when $k$=1, LaPerm shows significant degradation in its performance, which implies that one batch update is not enough to alter the structure of weights such that the newly added information can be effectively interpreted as permutations. In contrast, in (d) for $k$=2000, sharp fluctuations are observed after and before synchronization, which implies that the 2000 updates of the inner optimizer might have encoded information in a way that is beyond just permutations. However, we argue that this extra information in the latter case is not fundamentally important, as omitting it did not have a significant negative impact on the final accuracy.

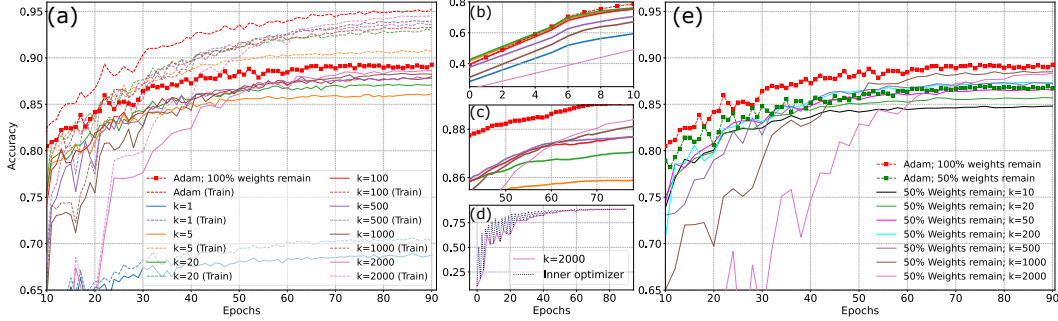

Figure 6: Conv4 on CIFAR-10. Conv4 trained for 90 epochs using a batch size of 50, with k chosen from {1, 5, 10, 20, 50, 100, 200, 500, 1000, 2000}. In (a)~(d), Conv4 is fully initialized. (b) and (c) are smoothed curves for specific epochs in the experiment. (d) shows validation accuracy of the inner optimizer when $k$=2000. (e) shows the behavior of LaPerm when the initial weights are randomly pruned. The training accuracies, if shown, are obtained by going through 30,000 randomly selected training examples.

Finally, we train Conv2, Conv4, and Conv13 initialized with $U_H$ on CIFAR-10 [23], using a batch size of 50, and compare LaPerm with *Adam* and LA. We safely choose $k$ to be 1000 for all architectures, i.e., the synchronization is done once per epoch.

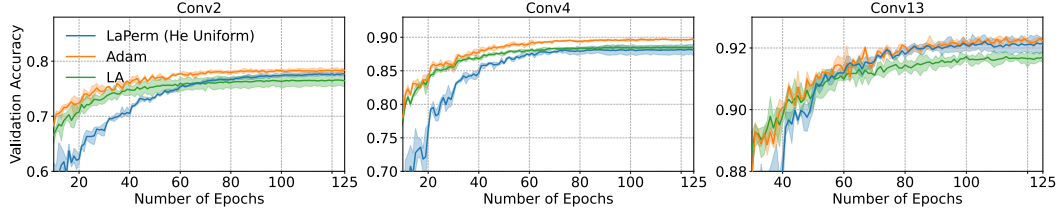

Figure 7: Validation accuracy of Conv2, Conv4, and Conv13 on CIFAR-10.

The results are shown in Figure 7. Similar to Figure 6, LaPerm using large $k$ started slow but demonstrated a steep growth trend in all cases. In addition, we observe growth in LaPerm's performance compared with regular optimizers as the network grows large. This observation could be partially attributed to a large number of possible permutations in heavily parameterized DNNs. For a DNN with $N$ weight vectors each of size $\#w_j$, not considering biases, LaPerm has access to $\prod_j^N \#w_j!$ different permutations. This number for Conv2, Conv4, Conv7, and Conv13 is approximately $10^{1.6e7}$, $10^{8e6}$, $10^{8e5}$, and $10^{5e7}$, respectively.

**5.3 Two Dimensions of Weights Hypothesis** We formalize our claim by hypothesizing that there are two dimensions of differences between initial and learned weights as follows. $D_1$: locations of weights; and $D_2$: the exact values of weights. $D_2$ can be further decoupled into a well-chosen common distribution ($D_{2_\theta}$) and the deviations of exact learned weights ($D_{2_\delta}$) from that distribution. Each weight vector $w_j$ can thus be represented as $w_j = \sigma_j(\theta) + \delta_j$, where $\theta$ is a vector drawn from the common distribution ($D_{2_\theta}$), $\sigma_j$ is a permutation operation ($D_1$), and $\delta_j$ is the remainder ($D_{2_\delta}$). However, this decomposition is not unique unless we put more restrictions on the choice of $\theta$ and $\sigma_j$. In section 2, by sorting the weight vectors (eliminating $D_1$), we observed SoWP in which a common distribution is enough to approximate all the information remaining in ($D_{2_\theta}$). SoWP implies that given a moderately chosen $D_{2_\theta}$, modifying $D_1$ alone can result in a performant DNN.

As demonstrated in Section 3, 5.1, and 5.2, SGD-based methods update $D_1$ and $D_2$ simultaneously at each learning step. However, we hypothesize that after enough iterations, the changes applied to $D_1$ and $D_2$ become increasingly isolatable as the SoWP begins to appear. Especially in Section 5.2, we demonstrate that LaPerm is more capable of extracting effective permutations when $k$ is larger.

In Figure 6(a), we observe a negative correlation between $k$ and the convergence speed in the early epochs. We consider it to be possible that although $D_1$ serves as a foundation for a trained DNN to perform well, its progress may or may not be immediately reflected in the performance. In Section 5.1, we saw how over-adapted (begin with a well-tuned $D_2$ before learning $D_1$) weight values $N_s$ performed poorly in the end. In Section 5.2, we saw that LaPerm with smaller $k$ converged faster, but to a worse final accuracy. We consider is to be possible that modifying $D_2$ can help the neural network

appear to learn quickly; however, without a properly established $D_1$, prematurely calibrating $D_2$ may introduce difficulties for further improving $D_1$. Nevertheless, LaPerm, which leaves $D_2$ as completely uncalibrated (the resulted weights are still random values), was usually slightly outperformed by Adam. It implies that $D_2$ might be crucial for the final squeeze of performance.

**5.4 Reconnecting Sparsely Connected Neural Networks** We further assess our hypothesis by creating a scenario in which $D_1$ is crucial: we let the neural network be *sparsely* connected with random weights, i.e., many weights are randomly set to zero before training. We expect a well-isolated $D_1$ to effectively reconnect the random weights and result in good performance.

We use $p$ to denote the percentage of initial weights that are randomly pruned, e.g., $p$=10% means that 90% of the weights in $\theta_0$ remain non-zero. We redo the experiments on Conv4 as in Figure 6 (a) with $p$=50%. As expected, in Figure 6 (e), we see that the removal of weights has no noticeable impact on the performance, especially when $k$ is large. In fact, the performance for $k$=1000 has improved.

Next, we create a scenario in which $D_2$ is crucial: we perform the same random pruning as in the previous scenario; while freezing all zero connections, we train the network using *Adam*. In the results shown in Figure 6 (e) labeled as "Adam;50% Weights remain", we observe its performance to be similar to that of LaPerm with $k$=50; it is clearly outperformed by LaPerm for $k \geq 200$. We consider the possibility that when $k$ is relatively small, it is difficult to trigger a zero and non-zero weight swap, as there are not enough accumulated updates to alter the rankings substantially. The resulting reconnection (permutation of weights within non-zero connections), in this situation, thus behaves similarly to SGD weight updates within frozen non-zero connections.

Since pruning 50% of the weights from an over-parameterized DNN may not significantly impact its accuracy, we test our hypothesis on ResNet50 [15], which has 760K trainable parameters (1/3 compared with Conv4), using a wider range of $p$. We initialize the network from $U_\mathrm{H}$. *Adam's* learning rates for all experiments begin at 0.001 and are divided by 10 at the 80th, 120th, 160th epoch, and by 2 at the 180th epoch. For ResNet50 with a different initial weight sparsity, we sweep over $k$ from $\{250, 400, 800\}$ and pick the one with the best performance.

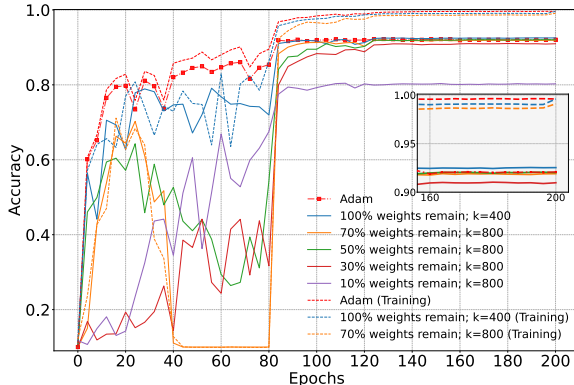

Figure 8: Accuracies of ResNet50 on CIFAR-10.

In the results shown in Figure 8, we observe the accuracies of LaPerm when $p \in \{0\%, 30\%, 50\%, 70\%\}$ to be comparable or better than that of *Adam* when $p$=0%, which again demonstrates the importance of a well-learned $D_1$. Moreover, before the first learning rate drop at the 80th epoch, LaPerm behaves in an extremely unstable manner. For $p$=30%, LaPerm acts as if it has diverged for almost 20 epochs, but when its learning rate drops by $10\times$, its training accuracy increases from 10% to 95% within three epochs. When $p$=50%, we observe a similar but less typical trend. This demonstrates that the progress on $D_1$ is not reflected in the accuracies.

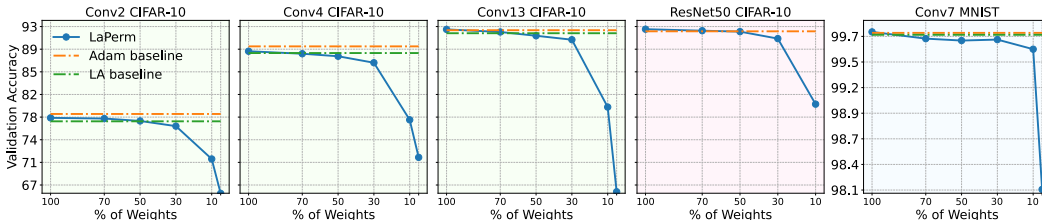

Figure 9: The networks are randomly pruned before training, and reconnected using LaPerm. The percentage of weights remaining in the network before training is shown as "% of Weights."

Finally, similar pruning experiments are performed on all architectures shown in Table 1. The results are summarized in Figure 9 using the previous hyperparameter settings. For all experiments, the weights are initialized from $U_\mathrm{H}$. We sweep over a sparse grid for $k$ from 200 to 1000 and safely choose the largest one that does not diverge within the first 50 epochs. We observe that LaPerm,

when $p \leq 70\%$, achieves comparable accuracy to that of the original unpruned network. Since setting weights to zeros in a weight vector decreases the number of permutations by a factor of the factorial of the number of zeros, we expect difficulties in optimization as the number of zeros increases. On the other hand, sparsely wired DNNs allow LaPerm, especially when $k$ is large, to trigger architectural (zero and non-zero weight swaps) change, which adds a new dimension to training that may neutralize the damage of losing the possible permutations. In addition, since LaPerm never alters $D_2$, they can be tuned to further encode information.

In all of our pruning experiments, *we simply remove $p\%$ of randomly chosen weights from every weight matrix in the network* except for the input layer which is pruned up to 20%. This naive pruning approach is not recommended if a high $p$ is the goal, as it may cause layer-collapse, i.e., improperly pruning layers hinders signal propagation and renders the resulting sparse network difficult to train [27]. In Figure 9, we observe a severe drop in performance when $p \geq 90\%$. The intent of using the most naive pruning approach is to showcase and isolate the effectiveness of a well-learned $D_1$. Since our work is on reconnecting the weights and is orthogonal to those on pruning the weights, previous works [2, 28, 33, 36, 46, 48, 49] can be combined to improve performance at higher $p$.

**5.5 Weight Agnostic Neural Networks** Inspired by what Gaier and Ha [10] achieved for weight agnostic networks, we reduce the information stored in $D_2$ to an extreme by setting all weights to a single shared value. We reconnect two simple networks: one with no hidden layers ($F_1$, a linear model) and one with two hidden ReLU layers of size 128 and 64 ($F_2$). Both networks use 10 softmax output units without bias. Since pruning is necessary for triggering architectural change, before training, we randomly prune 40% of the weights in $F_1$ and 90% of the weights in $F_2$. The remaining weights in $F_1$ and $F_2$ are all set to 0.08 and 0.03, respectively. We train $F_1$ and $F_2$ for 10 and 25 epochs, with a batch size of 128 using LaPerm with $k = 250$ on MNIST [26] for 30 random seeds each. As shown in Figure 10, we achieved ~85.5% and ~53% on $F_1$ and $F_2$. By using a slightly less naive pruning method, $F_2$ achieved 78.14% test accuracy. Detailed settings can be found in the Appendix A.5.0.

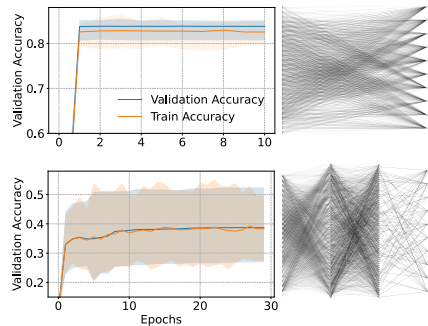

Figure 10: (left) Performance of LaPerm on weight agnostic networks trained on MNIST. (Right) The actual weight agnostic neural network obtained at the highest validation accuracy.

# 6 Related Work

**Lottery Ticket Hypothesis** Frankle el al. [9] explored a three-step iterative pruning strategy for finding random subnetworks that can be trained efficiently. They repeatedly trained the network, pruned $p\%$ of the weights, and reinitialized the remaining weights. For LaPerm, when $p\%$ of initial weights $\theta_0$ are pruned and $k$ is large, synchronization can be considered as finding the top $1 - p\%$ of weights in $\theta_t$ and "reinitializing" them. However, as opposed to reinitializing to their original values in $\theta_0$, as in [9], LaPerm sets them with *proper values* chosen from $\theta_0$ that match the learned rankings. We have demonstrated its virtues: LaPerm is performant despite being *one-shot* using only random weights, whereas the method in [9] requires heavy usage of the training data and to train from scratch for many times. In [9], the $p\%$ starts 0 and is gradually increased, whereas LaPerm shows promising performance even when the network is already randomly pruned before training by utilizing the relation between $D_1$ and $D_2$. We plan to explore varying $p$ during training for LaPerm in the future. Moreover, [9] conjectured that SGD seeks out and trains a subset of well-initialized weights. In Section 5.3, 5.4, 5.5, we empirically showed that it is crucial for SGD to seek out a good $D_1$ that might be pointing to the same direction. Our analysis might offer a complementary perspective for evaluating their conjecture.

**Random Weights and Super-masks** It was hypothesized and demonstrated in two recent works [53, 41] that pruning is training. In contrast, we show that reconnecting is training. For a vector of size $N$, they chose from $2^N$ different ways of masking, whereas we explore the space of $N!$ permutations. Ramanujan et al. [41] mentioned that they could not obtain desirable performance when the network is not sparse enough or too sparse. However, by exploiting the flexibility in reconnection, we are able to achieve promising results with a wide range of sparsity. Finally, it was interesting that the best validation accuracy when $p = 50\%$ on Conv2 and Conv4 described in [41] ~78% and ~86%

are similar to our results ~78% and ~88.5%. It is also intriguing that the better results in [41] were obtained using the *signed Kaiming constant*, as opposed to Kaiming's uniform [14] random values, as we observed.

**Learning Rate and Generalization** In Figure 6 (d), for Conv4 with LaPerm $k$=2000, a sharp increase in accuracy is observed between synchronizations. In Section 5.4, when training ResNet50 on Cifar-10, we encountered "resurrection" in training accuracy from 10% to 95% within three epochs after the learning rate drop. [34] investigated a phenomenon in which a small initial learning rate allows for faster training and better test performance initially, whereas large learning rate achieves better generalization soon after the learning rate is annealed. Their key insight is on how small learning rate behaves differently when encountering easy-to-generalize, hard-to-fit patterns v.s. hard-to-generalize, easier-to-fit patterns. Although we never use a large learning rate but forcefully and semi-randomly regularize the weights, our observation in Section 5 on the effect of sync period shares a similar philosophy with their insight. Drawing a connection between the magnitude of learning rate and the size sync period might offer an alternative perspective for understanding the generalization.

**Weight Agnostic Neural networks Search** Gaier and Ha [10] create neural networks that can perform various tasks without weight training. Different from [10], in Section 5.5, we did not create new connections or discriminate neurons by equipping them with different activation functions, but only reconnect basic layer-based neural networks. On MNIST, they created a neuron-based architecture using fewer connections but achieved better results (~92%) than ours (~85.5%). This may indicate an advantage of neuron-based architectures over basic layer-based ones.

# 7 Discussion and Future Work

This work explored the phenomenon of weights in SGD-based method trained neural networks that share strikingly similar statistical properties, which implies a surprisingly isolable relation between the values and the locations of weights. Exploiting this property, we proposed a method for training DNNs by reconnection, which has implications for both optimization and pruning. We presented a series of experiments based on our method and offered a hypothesis for explaining the results.

Is SoWP necessarily desirable? We have conducted preliminary experiments in which we force the violation of SoWP by initializing LaPerm with dissimilar weights. We observe that DNNs are still able to learn but show degraded performance. From a different direction, Martin et al. [37] analyzed the implicit self-regularization of DNNs during and after training by leveraging random matrix theory. Extending previous work on training dynamics [37, 13, 32, 20], future work might provide a theoretical explanation for SoWP.

Being able to train-by-reconnect would also simplify the design of physical neural networks [39] by replacing sophisticated memristive-based neuron connections [3, 44] with fixed-weight devices and permutation circuits. DNN trained by LaPerm can be reproduced as long as: (1) a rule is used to generate the initial weights; and (2) sets of distinct consecutive integers representing a permutation are presented. As a complementary approach to previous work on compression and quantization [47, 6], we can store the weights using Lehmer code [29] or integer compression methods [21, 30, 31].

In this work, we primarily utilized static hyperparameter settings. Future work could involve building an adaptive mechanism for tuning the pruning rate. Adjusting the sync period w.r.t. the learning rate would also be an interesting direction. Moreover, we have conducted experiments only for vision-centric tasks on small datasets (MNIST [26], CIFAR-10 [23]), we thus would like to contraint our claims to only classification problems. However, SoWP is also found in DNNs trained on larger datasets and different tasks, e.g., in trained GloVe [40] word embeddings. In future work, we plan to explore how DNNs learn when facing different and more challenging tasks.

LaPerm trains DNNs efficiently by only modifying $D_1$, but relies on a regular optimizer that applies changes simultaneously to $D_1$ and $D_2$ to lookahead. Can we create an optimizer that does not require the extra work done on $D_2$? In addition, LaPerm involves many random jumps and restarts while still able to train properly. We hope that our findings can benefit the understanding of DNN optimization and motivate the creation of new algorithms that can avoid the pitfalls of gradient-descent.

**Acknowledgements** We greatly appreciate the reviewers for the time and expertise they have invested in the reviews. In addtion, we would like to thank Vorapong Suppakitpaisarn, Farley Oliveira, and Khoa Tran for helpful comments on a preliminary version of this paper.

**Funding disclosure** The author(s) received no specific funding for this work.

## Broader Impact

The hardware implementation [8, 3, 44] can exploit the inherently distributed computation and memory components of DNNs. On the other hand, implementations of physical neural networks usually face difficulties when building a large number of neurons and weighted connections while guaranteeing its reconfigurability [39]. In this work, we presented a novel method that achieved promising results on a variety of convolutional neural architectures by reconnecting random weights. Consequently, for a DNN existing in the physical world to be made of a set of neurons, where each neuron owns a set of neuron connections that is made of different materials functioning as random weights, we can modify such a neural network to perform well for different image classification tasks by simply reconnecting its neurons. Our work might inspire alternative physical weight connection implementations. For example, we could replace sophisticated electrical adjustable weight devices with fixed weight devices and let a permutation circuit control the flow of the input to the weights. When the network needs to function differently, we could directly update the configuration of the permutation circuit without having to rebuild the network. This approach would potentially enable reconfigurable physical neural networks to be produced and deployed at a lower cost. However, such a system may be specifically vulnerable to common security risks associated with DNNs, such as adversarial attacks [12], as it might be difficult to regulate their usage and update them in a timely manner with improved adversarial robustness, as compared with their software-based counterpart. On the bright side, in addition to the obvious benefits, e.g., fast inference, that a physical neural network could offer, they could potentially be made into a new type of puzzle game or Lego®-like educational toy. This could benefit children, hobbyists, and experts who would like to tweak an physical artificial neural network to study how it works. In Section 5.5, and Figure 10, we have obtained a simple network $F_1$ made of around 3000 neuron connections with a single shared weight value, which is able to achieve over 85% accuracy using the MNIST dataset. They can be reconnected for classifying other data, e.g., Fashion-MNIST [50]. Since jigsaw puzzles on today's market come in sizes of 1000~40,000 pieces, assuming a sufficiently advanced technology in the future for producing permutation-based artificial neural networks, 3000 might not be an impressively large number.

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
