[Supplementary Material]

# A  Appendix

## A.1  Contents of Supplementary Materials

In addition to what is included in this Appendix, the supplementary material repository `https://github.com/ihsuy/Train-by-Reconnect` also includes the code and pre-trained weights. Detailed example usage of the code, e.g., training and validation script for reproducing the main results of the paper, are also included. A table of contents and the explanation for usage are included in `README.md` in the Supplementary Materials.

## A.2  Supplementary Materials for Section 2: Similarity of Weight Profiles

Figure 11: How to plot weight profiles. (a) Given weight vectors, (b) sort each weight vector in ascending order, (c) create a scatter plot for each weight vector, and (d) hide the *z*-axis. For definitions of terms, please refer to Section 2.

Complementary to Section 2, we present the weight profiles of pre-trained convolutional neural networks on ImageNet [7], including VGG16 [43], VGG19 [43], ResNet50 [15], ResNet101 [15], ResNet152 [15], ResNet152-V2 [16], DenseNet121 [18], DenseNet169 [18], DenseNet201 [18], Xception [5], NASNet-Mobile [54], and NASNet-Large [54]. The pre-trained weights of the afore-mentioned neural networks are downloaded directly from `keras.applciations` [4]. Since compiling all of the weight profile images into one file may harm the reading experience, we only show the weight profile of DenseNet121 here and store the rest of the images in a folder called `weight_profiles`, which is included in the Supplementary Materials.

Figure 12: DenseNet121. The image can be zoomed in for details.

Figure 13: Monitoring weight distributions, changes in the ranking distance (permutations), and validation loss in the first weight matrix of the network. Each column title indicates the experimental setting. Under each title from the top to bottom shows the evolution of: (Row 1) the weight distributions, shown as {15, 50, 85}th percentiles. Each percentile is displayed as 100 lines representing all 100 weight vectors. The 50th percentile is highlighted using red color. The maximum and minimum weights are shown as single lines above and under the percentiles, respectively. (Row 2) ratio of the mean and standard deviation of the ranking distance to the size of the weight vector, i.e., $\overline{D_t}/784$ and $SD[D_t]/784$. (Row 3) the trend of validation loss on 10,000 test images.

**Experimental Setting** We train a fully-connected DNN with two hidden layers (100 ReLU units each) and an output layer with 10 softmax units using the cross-entropy loss on MNIST [26]. The network is initialized with uniform random weights, as introduced in [14], and is trained using 60,000 training examples for 30 epochs with 50 examples per mini-batch. The network is validated on 10,000 test examples. The learning rates in all experiments are divided by 2 and 5 at the 10th and 20th epochs. In isolation, we train using the same architecture and initialization under four different settings: 1) SGD (initial learning rate: 1e-1) with no regularization; 2) *Adam* (initial learning rate: 1e-3) with no regularization; 3) *Adam* (initial learning rate: 1e-3) with 0.4 and 0.3 dropout [45] on the outputs of the first and second hidden layers; and 4) *Adam* (initial learning rate: 1e-3) with 2e-4 L2 regularization [24] on the weights associated with the first and second hidden layers.

**Analysis** We first study the evolutions of weight distributions in Figure 13 (a)~(d) and (i)~(l). The most noticeable distinction is spotted between (b) and (d), where in (b) the distribution expands but in (d), due to regularization, it collapses. On the other hand, (a) in comparison to (b) shows much less expansion. This is likely because the gradient updates of SGD tend to be scaled uniformly toward every dimension. A simplified example could be as follows. If we uniform-randomly update a uniformly distributed initial weight matrix for $n$ iterations, the resulting weight matrix would possess the properties of an Irwin–Hall distribution in which the standard deviation grows asymptotically to $\sqrt{n}$. Assuming that the updates in the actual training are sparse and their values are small, 30 epochs (with learning rate decay) would have a comparatively insignificant effect on the weight distribution. However, what we could learn from (a)~(d) is limited, e.g., despite demonstrating drastically different behaviors in validation losses in (j) and (k), (b) and (c) show subtle differences.

Next, we study the statistics of ranking distances in Figure 3(e)~(h). We observe that the mean distance, which is positively correlated with the total number of changes in ranking, might signify the intensity of learning. For example, overall, (h) maintains a larger but more fluctuating mean distance in comparison to (d), where the corresponding loss curve in (l) appears to be steeper and more unstable in comparison to (k). By contrast, when the network enters the phase where only a few permutations occur, we observe a flatter loss curve and milder fluctuations, signifying that the learning is nearly saturated, e.g., in (i) and (k) after the 20th epoch. Moreover, we notice that during such a saturated phase, any sharp jump in the mean distance could be a sign of overfitting. We consider the possibility that the network has encountered training examples that, according to its current knowledge, are outliers, despite that these examples were presented to the network many times in past epochs. Such sudden jumps could suggest that the network begins to fit the rest of the

examples too well. As a result, more permutations are triggered to cope with such outliers, which results in further overfitting. Parts (f) and (j) epitomize such a situation as follows. At around the 24th and 30th epochs, the sharp rise in the mean of the ranking distance predicts the deterioration in validation loss, without any knowledge about the validation data.

Moreover, the behaviors of permutations show unique traits under different settings. For setting (1), the trends seem highly random, especially at learning rate = 0.1. This is expected, because SGD uses the update rule $\theta \leftarrow \theta - \alpha \cdot \nabla_\theta L(\theta, d)$ for learning rate $\alpha$, weights $\theta$, and loss function $L$; thus, the updates occurring at each step are largely dependent on the randomly sampled batch $d$. In contrast, (f)~(h), i.e., the permutations caused by *Adam* with or without regularization, appear to be much less random. This might have to do with its update rule: $\theta \leftarrow \theta - \alpha \cdot m/\sqrt{v}$, where $m$ and $v$ are dependent on all previous gradients since the beginning of training; thus, given properly chosen hyperparameters, its behavior is not dominated by the randomness in training. Moreover, comparing (f) with (g), we see that dropout enables *Adam* to create, on average, larger and more stable-sized updates. Since dropout is equivalent to training different randomly sampled sub-networks, neurons are constantly placed in an environment where frequent self-correction is necessary. Finally, when L2 regularization is applied, the changes in ranking tend to be great in both number and size. This can be observed from (d) where the weight distribution collapses due to the L2 weight penalties, i.e., the weights, on average, become closer to each other. The closer two weights are from each other, the easier their rankings can be swapped and the larger the ranking distance the swap would cause by an update.

In conclusion, stochastic gradient-based optimizers not only permute the weights, we also observe frequent changes in the statistics of weights. Nevertheless, within these noisy fluctuations, we can distill substantial progress of learning by only looking at the relative ranking of the weights.

### A.4 Supplementary Materials for Section 4: Lookahead Permutation (LaPerm)

If necessary, we could accurately extract the permutations performed by LaPerm by directly comparing the rankings of weights between two consecutive synchronizations and deduce the permutations using a cycle-finding algorithm. Since permutation graphs are perfect, we could adopt simple algorithms, such as depth-first search (DFS), to efficiently find the permutations. A visualization of such permutations is shown in Figure 14.

A `TensorFlow` [1] implementation of LaPerm is included in the supplementary material. Please refer to A.1 for more details.

**The costs of lookahead.** Except when k is extremely small, LaPerm, on average, has few extra computational overheads in addition to the cost of its inner optimizer. For example, running the scripts provided in the supplementary material on a Google Colab GPU runtime, synchronizing Conv13 (14.9M parameters) once takes around 200ms. For Conv13 in Figure 6, we needed to synchronize totally 125 times which only added 25s to the overall training time. Moreover, a larger $k$ is observed to work well (e.g. Figure 6) and thus should often be used.

Figure 14: Permutations between the first two LaPerm (use Adam as inner optimizer, $k = 20$) iterations on a weight vector of size 128 in a convolutional neural network trained on the CIFAR-10 dataset. Vertices (black dots) representing the 128 weight values are aligned counterclockwise in a circle in ascending order. Each disjoint permutation cycle is marked using the same color.

### A.5 Supplementary Materials for Section 5: Experiments: A Train-by-Reconnect Approach

We describe extra experiment details that are not mentioned in Section 5. The complete visual-based architecture descriptions for all the neural networks used in this paper are included in the folder called `networks`. The train and evaluation scripts are also included in the supplementary material; please refer to A.1 for more details. Note that the accuracies for LaPerm for all experiments are calculated right after synchronization.

**A.5.0 Improve the Experiment Results**  The focus of our paper was not on pursuing state-of-the-art accuracy, but to gain an understanding of the effectiveness of a well-learned $D_1$, its relationship to $D_2$, and its possible implication on optimization and pruning. Therefore, we chose straightforward experimental settings for clear demonstrations. However, the experimental results described in Section 5 can be further improved if we refine the hyperparameters. We demonstrate this using the following examples.

For the last experiment in Section 5.4, we chose $k \le 1000$ from a sparse grid and obtained the results shown in Figure 9. However, better values of $k$ exist, e.g., when $k$=2000 (using the same hyperparameter settings), as demonstrated in Figure 15, we are able to achieve a better result compared with what was mentioned in Section 5.4. We expect that a fine-tuned $k$ or a schedule designed for $k$ can further improve the performance of LaPerm.

In Section 5.5, we used the same pruning rate for all three weight matrices of $F_2$ (hyperparameter details in Appendix A.6.4). However, since there are 100352, 8192, and 640 parameters in the weight matrices, respectively, a simple method for improving the pruning without introducing additional complexity would be to prune while considering the number of parameters, e.g., heavily parameterized matrices should be pruned more. We reconduct the experiment and randomly prune the three weight matrices of $F_2$

Figure 15: Randomly prune Conv4 and reconnect it using LaPerm with $k$=1000 and 2000. The percentage of weights remaining is indicated by "% of Weights".

at rates of 93%, 86%, and 67%, respectively (7%, 14%, and 33% of weights remain nonzero). We achieved a test accuracy of 78.14%, which is much higher than the result mentioned in 5.5, i.e., ~53%. Note that the results of all other pruning experiments, e.g., in Section 5.4, can be potentially improved by taking into account the size of weight vectors while setting the pruning rate, as opposed to using the same pruning rate for all layers.

**A.5.1 General Information about the Datasets**  In this paper, we considered classifying images using the MNIST [26] and CIFAR-10 [23] datasets. The MNIST dataset consists of 70,000 black-and-white images of size $28 \times 28$ with 10 different categories. The CIFAR-10 dataset consists of 60,000 colored images of size $32 \times 32$, with 10 different categories.

**A.5.2 Experiment Details for Section 5.1 Varying the Initial Weights**  For MNIST, we normalize both training and test data and use real-time random data augmentation with a rotation of up to 10 degrees, width and height shifts of up to 10% of the original image size for the training data, and random zoom at a range of 10%. The learning rate for *Adam* (both as an individual optimizer and inner optimizer) starts with 1e-3 and is multiplied by 0.95 at the end of each epoch. For LA, we use the TensorFlow [1] default settings, i.e., sync period 6 and slow step size 0.5. The networks are trained on 60,000 sample images and validated on 10,000 test images. For Conv7, no regularization except for dropout [45] is used.

**A.5.3 Experiment Details for Section 5.2 Understanding the Sync Period k**  For CIFAR-10, we z-score normalize (subtract by mean and divide by standard deviation) all images, and use real-time random data augmentation with rotation up to 15 degrees and width and height shifts of up to 10% of the original image size and random horizontal flip. The networks are trained on 50,000 training images and validated on 10,000 test images. For all experiments on Conv2, Conv4, and Conv13 in this section, we use a L2 regularization of rate 1e-4, dropout[45], and BN [19]. The BN [19] layers are updated regularly using the inner optimizer of LaPerm. *Adam* (both as an individual optimizer and inner optimizer) uses an initial learning rate of 1e-3, and is multiplied by 0.6 at every 10th epoch.

In addition, LaPerm appears to need repeated synchronizations to find the optimal reordering. We conducted experiments on Conv4 under the same setting as in Section 5.2, but choose to synchronize only once at the end of the training ($k = 90000$), we obtained on average 13.8% (both validation and training) accuracies.

**A.5.4 Experiment Details for Section 5.4 Reconnecting Sparsely Connected Neural Networks**
We apply the same data preprocessing, data augmentation, and train-validation split as in the previous section. For Conv2, Conv4, and Conv13, we use the same regularizations and training hyperparameters as in the previous section. For ResNet50, the learning rates of *Adam* (both as an individual optimizer and inner optimizer) begin at 1e-3 and are divided by 10 at the 80, 120, 160th epoch, and by 2 at the 180th epoch.

Since the input layer usually has significantly fewer weights in each weight vector, to avoid creating bottlenecks, the input layer is always pruned only up to 20% (at most 20% of weights are set to zero), whereas the remaining layers share the same rate of pruning as described previously. The BN layers and biases (Conv7) are not pruned.

**A.5.5 Experiment Details for Section 5.5 Weight Agnostic Neural Networks** For the experiments in the section, we perform the same data normalization as mentioned in A.4.1, but do not use data augmentation. For both experiments, we use an initial learning rate of 1e-3 and multiply it by 0.95 at each epoch. For $F_1$, we do not use regularization. For $F_2$, we use L2 regularization of rate 1e-4 on the hidden layers. The weight matrix of $F_1$ is randomly pruned by 40% (40% of weights are set to zeros). The weight matrices of $F_2$ are randomly pruned by 90%.

**A.5.6 Usage of Batch Normalization** As mentioned in A.5.3, we used BN [19] in Conv2, Conv4, and Conv13. In Section 5.4, we follow the original design of ResNet [16] and thus also adopt BN. The BN layers in the aforementioned experiments are updated regularly using the inner optimizer of LaPerm, i.e., they are not permuted and set to random values. Our intent is to use BN as an optimization tool.

However, the usage of BN may create concern in regard to where the information is actually located, i.e., one could completely attribute LaPerm's effectiveness to BN's learned scaling ($\gamma$) and shifting ($\beta$) terms. On the other hand, removing BN from all the aforementioned architectures may render the networks difficult to train, and we cannot obtain results comparable to those of related works under similar settings.

To resolve this dilemma, we propose the following "$\gamma\beta$ *reset*" training scheme to isolate the contribution of $\gamma$ and $\beta$ from LaPerm-trained DNNs. Since ResNet50 uses the highest number of BN layers among the chosen architectures, we use it as an example to demonstrate our point. We use BN

Figure 16: $\gamma\beta$ reset experiments.

as usual in ResNet50 and update $\gamma$ and $\beta$ using the inner optimizer of LaPerm. *However, at each synchronization, we reset $\gamma$ and $\beta$ to 1 and 0.* We compare its performance with LaPerm (never reset $\gamma$ and $\beta$) using both $k = 800$ (other experimental settings are the same as in Section 5.4). We repeat the experiment three times and show the results in Figure 16. We observe only roughly a 1% decrease in the final accuracies when $\gamma$ and $\beta$ do not hold information. Note that the difference demonstrated in Figure 16 is similar to that between training ResNet50 using a regular optimizer with and without BN [51]. The proposed experiment demonstrates the effectiveness of LaPerm as the main horsepower for training.