[Reviews · NeurIPS 2020]

Review 1

Summary and Contributions: This paper proposes an approach to train neural networks by permuting randomly-initialized weights. It introduces an algorithm called Lookahead Permutation (LAPerm) that works as follows: 1. The weights of a neural net are randomly initialized 2. The random weights are sorted by value 3. Optimization is performed for k steps as usual *on a copy of the model*, using a standard gradient-based optimizer like SGD or Adam 4. After every k optimization steps, the resulting weights are sorted by value to form a ranking, and then these weights are overwritten by the corresponding rank entries from the sorted random initialization. This process, termed "synchronization," restricts the weights to the set of random initial weights. The main hyperparameter introduced by LAPerm is the synchronization period, k, which can have a significant impact on training. The paper evaluates LAPerm on three tasks: 1) training standard CNN classifiers on MNIST and CIFAR-10 by permuting random weights; 2) training sparsely-initialized networks from scratch on CIFAR-10; and 3) training Weight Agnostic Neural Networks on MNIST (e.g., sparse networks in which all the parameters have the same fixed value).

Strengths: * This paper empirically demonstrates that randomly initialized weights, properly permuted, can achieve reasonably good performance on MNIST and CIFAR-10. * LAPerm is able to train sparse networks with ~30% of the original number of weights to reasonable accuracies, only slightly degraded from the performance of all weights. * The ranking of a weight vector is clearly described in Section 3. * Figure 14 in Appendix A.4 is a neat illustration of the effect of permuting random weights for an MNIST logistic regression model. It might be nice to move this figure to the main paper. * The conclusion presents an interesting potential use-case of permutations, to store/transmit neural networks in a compressed representation where the only pieces of information needed to re-create the network are: 1) the seed for a pseudo-random number generator of a specific type of noise; and 2) a permutation expressed by a list of integers. - This is conceptually interesting for lightweight storage and transmission, especially for dense network weights. Unfortunately, it is not an alternative to pruning because the reconstructed networks would still be memory-intensive on-device and expensive to use at test-time.

Weaknesses: * I am not sure how novel or meaningful the analysis of "weight profiles" is in Section 2. Checking the provided code, the weight profiles in Figures 1 and 2 are plotted for the weights in an ImageNet-pretrained model as: vgg16 = tf.keras.applications.vgg16.VGG16(include_top=True, weights="imagenet") It would be important to know what hyperparameters were used in the training script for the pre-trained models. It is likely that the weight initialization was Gaussian, and that weight decay was used for regularization. Then the distribution of weights in the trained model may not differ too greatly from the initial distribution (e.g., still roughly Gaussian). One can obtain similar plots to Figure 2 by sorting random Gaussian samples: samples = np.random.normal(size=(1000,)) plt.scatter(np.arange(len(samples)), np.sort(samples)) Alternatively, there are many distributions other than Gaussians that could potentially yield similar heavy-tailed plots as Figures 1 and 2. A relevant paper looking at the distributions of trained network weights is [1]. [1] Mahoney & Martin, "Traditional and Heavy Tailed Self Regularization in Neural Network Models." ICML 2019. * I think this paper requires more precise writing, including mathematical statements formalizing the claims. Currently there are many parts of the paper that are vague and not explained in sufficient detail, in particular Section 5.2 on the "Two Dimensions of Weights Hypothesis" where D_1 and D_2 are discussed. This section has many informal arguments. - "Within a few steps of SGD, an iterative process where each step contributes a small amount toward the objective, the changes are simultaneously applied to both D_1 and D_2. However, as these changes accumulate, D_1 and D_2 tend to become increasingly perpendicular." --> Statements like "become increasingly perpendicular" are informal and may confuse the reader more than they elucidate. * Appendix A.5.3 states that BatchNorm is used in the CIFAR-10 CNN experiments. BatchNorm has internal statistics updated during training, and parameters that can yield decent performance even on top of random networks [2]. How is BatchNorm handled when using LAPerm, e.g., what happens to the statistics and learnable shift/scale parameters? [2] Frankle et al. "Training BatchNorm and Only BatchNorm: On the Expressive Power of Random Features in CNNs." arXiv 2003.00152, 2020. * From my perspective, the proposed method can be understood as a regularizer that "snaps back" arbitrary parameter values to be equal to a specific ordering of the initial parameters. In this sense, it is somewhat similar to quantization. The fact that LAPerm restricts the values of weights in this way intuitively prevents overfitting. In this vein, I think the paper should more clearly show and discuss training loss/accuracy: training performance is not shown in Figures 4, 6, and 8. * LAPerm is only applied using Adam as the inner optimizer; it would be good to demonstrate that the algorithm is robust to the choice of inner optimizer. Does using a different optimizer affect sensitivity to the sync period k? * I think the writing is vague on lines 181-182: "implies the 2000 updates of inner optimizer might have encoded an amount of information that beyond just permutations." * As a regularizer, LAPerm is fairly sensitive to the sync period k. * What if you initialize with low-precision weights? Or binary 0-1 weights? Does LAPerm allow one to learn such networks? Would the sync period be even more critical for these tasks? * If it is correct to say that LAPerm enforces that the distribution of the weights stays the same throughout training---e.g., if the initialization is Gaussian, then the final trained (that is, permuted) weights will also be Gaussian-distributed---then this adds to the regularization interpretation of the method. * Lines 23-25: I do not think it is a good idea to describe the training process for deep neural nets as a "black box." This disregards a lot of work in recent years towards understanding neural net training dynamics, including the importance of the initial phase of training [3] and theoretical analysis of infinite-width networks [7] (this reference is given in the "Relation to prior work" section). [3] Frankle et al., "The Early Phase of Neural Network Training." ICLR 2020.

Correctness: * Most of the experimental setups and comparisons seem correct. * Several of the claims are too vague to be evaluated for correctness. - The paper lacks necessary mathematical formalism, and lacks theoretical justification for its algorithm and analyses.

Clarity: * Overall, the paper does not seem focused enough. It is unclear whether it aims to analyze a phenomenon regarding the distribution of weights before/after training, or investigate the properties of permutation-based optimization, or propose a new algorithm for sparse network training. Sections 2 and 3 seem to slant towards analysis, but the following sections focus on introducing a new algorithm and trying to show that it is practical and performs well. The experiments section is also confusing regarding the main point of the method: is it a regularizer, or a new way to train sparse networks, or a way to train weight-agnostic networks? The experiments are fairly small-scale, so it is unclear whether LAPerm scales successfully to harder or more realistic tasks (CIFAR-100, ImageNet). * The organization of the Related Work is hard to follow, because it mixes discussions of the related papers with empirical comparisons to some related methods. In the paragraphs on "Random Weights and Super-Masks" and "Weight Agnostic Neural Networks Search," results are compared between those papers and LAPerm. These comparisons seem out of place here, and should be moved to the experiments section. (For example, Figure 9 should be in Section 5.5, otherwise it feels disconnected.) * When Lookahead Permutation is introduced at the start of Section 4, it says that the algorithm is inspired by the Lookahead optimizer and Reptile optimizer, but the connections to them are not discussed. It is unclear what the significance of the relationship to LA is; it seems the connection is loose, with the similarities being that both LA and LAPerm involve an inner optimization loop and a "synchronization" step (that does something different in each algorithm). * The LAPerm algorithm is clearly presented. * Each of the experiments uses a different CNN architecture or set of architectures. Figure 4 uses Conv7; Figure 5 uses Conv4; Figure 6 uses Conv2, Conv4, and Conv13; Figure 7 uses ResNet50; Figure 8 uses Conv2, Conv4, Conv7, Conv13, and ResNet50. Why are these arbitrary choices made? * Line 75: "Adam (1e-3) with L2 weight decay and learning rate drops." - It is not clear whether all the results in Section 3 use learning rate decay, or if it is only used by this particular run (Adam + weight decay). Appendix A.3 suggests that all the runs use LR decay, so this should be clarified. - Note that L2 regularization and weight decay are not equivalent when using Adam [4]. Could the authors clarify which regularizer they are using, and correct the terminology "L2 weight decay" used in Figure 3? [4] Loshchilov & Hutter. "Decoupled Weight Decay Regularization." ICLR 2019. * Figures 5 and 7 are very difficult to read: they have extremely small legend text, too many curves plotted simultaneously, and tiny inset boxes. In addition, subplots (c), (d), and (e) in Figure 5 are *very* small. All of these figures require maximum zoom levels to read. * Table 1 containing architecture details is not crucial for the main text, and can be moved to the appendix. * The caption of Table 1 mentions "ConvSmall" that does not appear in the table. * In Figure 4, "Glorot Normal", "Shuffled pre-trained", "Adam", and "LA" are hard to tell apart because the colors are too similar. * In Section 5.1, there is no reason to introduce the shorthand U_H, N_H, U_G, N_G, and N_S for the different normalizations, because these abbreviations are never used (not even in the legend of Figure 4). * Line 259: "one without hidden layer" - Is this referring to a linear model? * Lines 91-92: "since Adam's updates take into account previous gradients, the permutation appears to follow a certain trend" - I think this conjecture should be verified by also including an experiment using SGD+momentum, which is simpler than Adam and also takes into account previous gradients. * Line 106: "\theta_{sorted}, which is updated regularly ..." - This is gramatically confusing because it is actually \theta_t that is updated regularly. * Line 278: "In 5, we ..." --> "In Section 5, we ..." * Line 13: "fully-initialized network" --> I believe this should be "dense network" * Line 30: I think this should be \theta_T = \sigma_T(\cdots (\sigma_1(\theta_0))). That is, the outermost (e.g., final) permutation should be denoted \sigma_T, not \sigma_1 as is currently written. * Some parts of the writing could benefit from a grammar/wording check. - Line 32: "guaranteeing the location of weights" --> it is unclear what is meant by "guaranteeing" * Lines 36-37: 'We show that stochastic gradient updates can be translated into "permutations."' - I think this should be expressed more precisely in the introduction, currently it is not clear why "permutations" is in quotes. Minor ----- * Figure 3 does not have x-axis labels. * Line 169: There is probably a typo when referencing Section 5.2, it should probably be Section 5.1.

Relation to Prior Work: * This paper does a reasonably good job of citing relevant prior work, including the lottery ticket hypothesis and weight-agnostic neural networks. * However, I think there is some missing discussion on a few related areas, that would help to clarify aspects of the paper and better position it in the literature. Here are a some citations in each of these areas: Neural network training dynamics -------------------------------- [5] Gur-Ari et al., "Gradient Descent Happens in a Tiny Subspace." arXiv 1812.04754, 2018. [6] Li et al., "Measuring the Intrinsic Dimension of Objective Landscapes." ICLR 2018. [7] Jacot et al., "Neural Tangent Kernel: Convergence and Generalization in Neural Networks." NeurIPS 2018. Networks with random weights and/or random connections ------------------------------------------------------ [8] Saxe et al., "On random weights and unsupervised feature learning." ICML 2011. [9] Du et al., "Gradient Descent Provably Optimizes Over-parameterized Neural Networks." ICLR 2019. [10] Zhang et al., "Are All Layers Created Equal?" arXiv 1902.01996v3, 2019. [11] Yehudai & Shamir. "On the Power and Limitations of Random Features for Understanding Neural Networks." NeurIPS 2019. [12] Rahimi & Recht. "Random Features for Large-Scale Kernel Machines." NeurIPS 2007. [13] Rahimi & Recht. "Weighted Sums of Random Kitchen Sinks: Replacing Minimization with Randomization in Learning." NeurIPS 2008. [14] Xie et al., "Exploring Randomly Wired Neural Networks for Image Recognition." ICCV 2019. Neural network compression and quantization ------------------------------------------- [15] Ullrich et al., "Soft Weight-Sharing for Neural Network Compression." ICLR 2017. [16] Hubara et al., "Binarized Neural Networks." NeurIPS 2016.

Reproducibility: Yes

Additional Feedback: Please see the other sections for comments, suggestions, and questions. Update after rebuttal: ---------------------------- I have read the other reviews and the authors' rebuttal. I kept my score of 5 unchanged. The authors have addressed some of the issues I raised, but many more remain. I thank the authors for providing a figure addressing my BatchNorm question. However, the authors did not address my concern regarding the theoretical justification of the method. It would be important to address this in the paper at hand, not in a follow-up one (as the authors state in their "Response to details and clarity"). The authors' response in bullet 3 of the rebuttal attempts to clarify the "components" of learned weights, but this would need to be expanded substantially for the paper (the proposed decomposition w=sigma(theta)+delta just assigns variables to quantities that are still defined in a vague way). I am not fully convinced by the authors' response about weight profiles, because all the weight profile visualizations shown in the paper and supplementary material are for pre-trained ImageNet models. It is not clear from the rebuttal how using different initializations (e.g. uniform) leads to weights that are "still roughly Gaussian with SoWP." Several important conceptual aspects are not sufficiently addressed by the rebuttal, including how we can interpret the regularization effects of LAPerm (where the authors claim that training by permutation implies more than just regularization). The questions related to the focus of the paper, the use of different networks in different figures, and the organization of the paper have not been adequately addressed. The authors' comments indicate that they are also not going to investigate LAPerm in larger-scale experiments, leaving that for the future too. As the authors clearly summarize in the last paragraph of their rebuttal, there is a lot of work to be done to revise the organization, writing, and figures of this paper. I would not feel comfortable accepting this paper without further review of those changes, so I do not think this paper is ready for publication at this time.


Review 2

Summary and Contributions: This paper empirically studies the dynamics of stochastic gradient descent optimizers which are the most used optimizers for deep learning. They basically divide the parameter learning process into two perspectives: weight value updating and weight ordering (the relative rank of the values of the weights). Throughout experiments using image datasets like CIFAR and MNIST, they show that weight ordering is very important for obtaining a performant neural network. In most experiments, by just swapping the order of randomly initialized weights, they are able to perform as well as if the weights are updated thoroughly by these optimizers. They also evaluated many hyperparameters on this process like how to initialize the weights, the interval to compute weight permutations, and other factors that give insight to the learning process. Finally, they evaluated their technique in the network pruning reaching very good results.

Strengths: The work provides very interesting insights into the most used optimizers for deep learning which is very useful for the community and can be developed further in future works. Their experiments used different network architecture, datasets, and hyperparameters which increases the robustness of their observations. The observation about the importance of the ranking of the weights to learn a neural network can be an inspiration for new optimization or regularization methods for deep learning and consequently future works.

Weaknesses: The experiments performed by the authors make sense in classification problems where the decisions are made by comparison of the outputs. But, how does it translates to regression problems? In regression problems, I think some of the paper observations would change since the exact value of the output is necessary. In my opinion, it is not a big problem. However, it would be good to constraint the claims in this paper to classification problems or provide some reason why it can be generalized for regression since it is not trivial. The paper only focus on quantitative analysis using classification metrics. They do not analyze the difference between the predictions on different samples when reconnecting the weights. Or the difference in the learned features and classification decision boundary. What is the reason to have multiple synchronizations in LaPerm? If the ordering of the weights is what matters, why not compute the permutation of the initial weights to the final weights directly?

Correctness: I have not identified problems in the formulation or in the experimental protocol. I have only a small question about how the optimizers are configured for a fair comparison. Are the models cross-validated (learning rate, weight decay, and learning rate schedule at least) for comparison? The batch schedule is controlled? How many repetitions are performed?

Clarity: The paper reads well and is easy to follow.

Relation to Prior Work: The related work section is good in general.

Reproducibility: Yes

Additional Feedback: >> After authors feedback: The authors have promised to limit their claims to classification problems and explained why the proposed algorithm needs multiple synchronization steps. However, they did not answer my questions about the details of the experiments as described in the correctness section above. Then, I decided to keep my original score since I still have some doubts about the robustness of their experiments.


Review 3

Summary and Contributions: The paper discovers a very interesting phenomenon: only reordering the initial weights in a neural network can achieve on-par accuracy (tested on small datasets) without changing the weight values. The reordering is still guided by an inner standard SGD optimizer (like Adam). This interesting finding is inspiring and sufficient for an acceptance.

Strengths: 1. a new finding that initial weights in a neural network don't need to be trained, but a simple ordering of those initial weights can achieve good generalization. 2. a set of comprehensive experiments to demonstrate and evaluate the hypothesis.

Weaknesses: 1. the method still relies on an inner standard SGD optimizer to find the reordering; 2. no large-scale experiments (ImageNet) are done. 3. demonstrating the performance under a higher sparsity (e.g. in Line 222 and Figure 7) will be more interesting.

Correctness: Correct.

Clarity: 1. In Line 30, the subscripts (1...T) should be reversed. _x001B_\delta_k can be misunderstood as the permutation at step k. 2. There is a missing link between the observation Figure 1&2 and the "Train by Reconnect". The Figure 1&2 illustrate the profile between different weight vectors, but the "Train by Reconnect" refers to a profile for one weight vector across the whole training steps. The former is a spatial comparison, while the latter is a temporal comparison; 3. In Line 153, "Adam and LA in isolation", cite Adam and LA. 4. "For Conv13, LaPerm outperforms Adam at its best validation accuracy": use specific numbers. Their curves look the same.

Relation to Prior Work: Include related work on neural network pruning and sparsifying below: Han, Song, Jeff Pool, John Tran, and William Dally. "Learning both weights and connections for efficient neural network." In Advances in neural information processing systems, pp. 1135-1143. 2015. Wen, Wei, Chunpeng Wu, Yandan Wang, Yiran Chen, and Hai Li. "Learning structured sparsity in deep neural networks." In Advances in neural information processing systems, pp. 2074-2082. 2016. Li, Hao, Asim Kadav, Igor Durdanovic, Hanan Samet, and Hans Peter Graf. "Pruning filters for efficient convnets." arXiv preprint arXiv:1608.08710 (2016). Bellec, Guillaume, David Kappel, Wolfgang Maass, and Robert Legenstein. "Deep rewiring: Training very sparse deep networks." arXiv preprint arXiv:1711.05136 (2017).

Reproducibility: Yes

Additional Feedback: 1. the reference of sub-figures of Figure 1 is wrong in a few lines (e.g. Line 56) and in the caption of Figure 2. 2. "For setting (1), the trends seem random": it converges quite well. 3. In Line 125, clarify k|N. 4. “implmentations of” -> implementations of ============ Rate remains as rebuttal provides little feedback to me (since I gave a good rate originally)


Review 4

Summary and Contributions: The paper studies the similarity of weight profile phenomena of trained neural networks and propose a lookahead permutation algorithm that only modifies the neural connections while keeping the weight values unchanged. Under thorough empirical studies, the proposed algorithm is demonstrated to achieve good performance, and can also be used as a pruning tehcnique. The paper also discusses potential applications and relashionships with fields such as winning ticket lottery hypothesis and weight agnostic neural network architecture search.

Strengths: The idea to only modify the connections of neurons is novel. The proposed algorithm is simple but effective as demonstrated in empirical results. The aspect of studying the neural network is inspiring and can be connected to many other lines of works such as winning-ticket lottery hypothesis and network weight pruning. The paper is purely empirical and supports its claim with thorough experiments, making the results quite convincing.

Weaknesses: This work is purely empirical based on the similarity of weight profiles observations. The paper lacks theoretical justifications/explanations of why SoWP is observed. Moreover, it would be nice if the paper can connect theoretically to rescent results from winning ticket lottery hypothesis. The proposed algorithm seems not quite useful in practice because of the costs of the lookahead operations. Maybe it's worth considering less costly lookahead operations or other ways to update the permutations to make the algorithm more efficient.

Correctness: They seem to be correct.

Clarity: The presentation is very clear.

Relation to Prior Work: The relashionships to previous works are well discussed.

Reproducibility: Yes

Additional Feedback: After authors' feedback: The authors provide timing results on my concerns with the lookahead costs. Given the numbers, the additional lookahead costs seem acceptable. I concern about lacking of theoretical connections remain. After reading other reviewers' comments and the corresponding feedbacks, I agree that there is room to improve the paper's clarity. The question on the effects of batchnorm training is a good question and I think the authors provide satisfying answers.

[Author Response · NeurIPS 2020]

We greatly appreciate the reviewers for the time and expertise they have invested in the reviews. This essay focuses on addressing the main concerns raised by each reviewer. Due to the page limit, we will not be able to mention and answer all the questions. However, we would like to thank the reviewers for every observation and suggestions for revision and clarification and will make sure to improve on every aspect of our paper.

**1. The meaningfulness of the analysis of "weight profiles". (Reviewer 1)** In Section 2, our focus is on the statistical similarity between weight vectors within a trained weight matrix and it is crucial for later developing the insight: reordering of neuron connections can result in learning. A majority of the main results in our paper are obtained using *uniform random distribution* as opposed to the gaussian-like distribution seen in well trained neural networks. Moreover, we have conducted experiments (not mentioned in the paper) using different initializations, e.g., uniform, with different scales and observed that the resulting weights to be still roughly gaussian with SoWP.

**2. Training BatchNorm (BN) alone yields decent performance [1]. (Reviewer 1)** To isolate the contribution of BN's scaling ($\gamma$) and shifting ($\beta$) terms from LaPerm reconnected networks, we propose to add experiments using the following "$\gamma\beta$ *reset*" training scheme into the appendix. Take ResNet50 as an example: we use BN as usual and update $\gamma$ and $\beta$ using the inner optimizer of LaPerm. At each synchronization, we reset $\gamma$ and $\beta$ to 1 and 0. We compare its performance with LaPerm (never reset $\gamma$ and $\beta$) both using $k = 800$ (other experiment settings are the same as Section 5.4). We repeat the experiment three times and show their results on the right. We observe only roughly 1% decrease in the final accuracies when $\gamma$ and $\beta$ do not hold information. In contrast, [1] focuses on the expressive power of $\gamma$ and $\beta$.

**3. Section 5.2 lacks sufficient detail. (Reviewer 1)** We would like to formalize our claim as following: We hypothesize that there are 2+1 dimensions of differences between initial and learned weights: $D_0$: Common distribution of weights, $D_1$: Locations of weights, and $D_2$: Deviations of exact learned weights from the best weights representable by well-chosen distribution ($D_0$) and locations ($D_1$). Each weight vector $w_j$ is represented as $w_j = \sigma_j(\theta) + \delta_j$, where $\theta$ is a vector drawn from the common distribution ($D_0$), $\sigma_j$ is a permutation operation ($D_1$), and $\delta_j$ is the remainder ($D_2$). This decomposition, however, is not unique, unless we put more restrictions on the choice of $\theta$ and $\sigma_j$. — The rest of the subsection will be revised accordingly.

**4. LaPerm can be understood as a regularizer and is sensitive to $k$. (Reviewer 1)** We agree that LAPerm has inherent regularization effects. But our claim — DNNs can be trained by only reconnecting random weights — would imply more than just regularization. Sections 5.4 and 5.5 provide partial results supporting our view. As for sensitivity of $k$, if we define being sensitive as "responding unpredictably to slight changes", then LaPerm is not quite sensitive. For example, we showed in Section 5.2 that the performance of LaPerm responded monotonically w.r.t. $k$ and changing it from 500 to 2000 effects only 2% of the accuracies.

**5. The paper does not seem focused enough. (Reviewer 1)** Our focus is dual: (1) most of the information of DNNs is stored in the orderings of weights, (2) we can actually construct algorithms to find good orderings. We think both are equally important and cannot defocus one of them. We would like to improve the first paragraph of sections 2 to 5 in the final version of the paper to further emphasize this structure for clarity.

**6. What is the reason to have multiple synchronizations in LaPerm? (Reviewer 2)** LaPerm appears to need repeated synchronizations to find the optimal reordering. We conducted experiments on Conv4 under the same setting as in Section 5.2, but choose to synchronize only once at the end of the training, we obtained on average 13.8% (both validation and training) accuracies. We will merge this experiment into Figure 5 to stress this point.

**7. Performance under a higher sparsity. (Reviewer 3)** Results for higher sparsity for Conv4 on line 222 are reported in Figure 8 (2). Instead of only showing their final performance w.r.t. "% of weights" as we did in the paper, we will add their detailed training trend in the appendix. In Figure 7, we will show two more trends for $p = 5\%$ and $1\%$.

**8. The costs of lookahead. (Reviewer 4)** Except when k is extremely small, LaPerm, on average, has few extra computational overheads in addition to the cost of its inner optimizer. For example, running the scripts provided in the supplementary material on a Google Colab GPU runtime, synchronizing Conv13 (14.9M parameters) once takes around 200ms. For Conv13 in Figure 6, we needed to synchronize totally 125 times which only added 25s to the overall training time. Moreover, a larger $k$ is observed to work well (e.g. Figure 5 in the paper) and thus should often be used. We will improve the main text in Section 4 to stress this point.

**9. Response to details and clarity. (All Reviewers)** We thank all the reviewers for the careful observations. We will improve the clarity of the figures. We will revise the main text and expand the paper's references and appendix according to the reviewers' suggestions in their "Correctness", "Clarity", and "Prior work" sections, especially focus on constraining our claims to only classification problems, stressing the architecture choices and their purposes, and fixing errors such as changing L2 weight decay to L2 regularization, fixing ambiguity and mistakes in notations and formulas. Due to limited time and space, we are unable to answer the comments about: trying other inner optimizers, conducting large-scale experiments, analyzing decision boundaries, theoretical justifications. We will pursue them in future works.

[1] Frankle et al. "Training BatchNorm and Only BatchNorm..." arXiv 2003.00152, 2020.

[Meta-Review · NeurIPS 2020]

This paper presents a series of analyses to disentangle the role of the locations of weights from their values. By training networks to mirror locations, but not values, the paper shows that networks can be trained to good performance. All the reviewers found the work to be interesting and novel, though there were some concerns about clarity/focus. While I agree that there are some clear ways to improve the clarity, I think the paper has enough strengths to merit acceptance. However, I do encourage the authors to take the feedback regarding clarity to heart and focus on improving this for the camera-ready version. The paper will benefit in the long run from some additional work in this area.